# Strategies for developing sustainable health research capacity in low and middle-income countries: a prospective, qualitative study investigating the barriers and enablers to locally led clinical trial conduct in Ethiopia, Cameroon and Sri Lanka

Samuel R P Franzen,[1,2] Clare Chandler,[3] Sisira Siribaddana,[4] Julius Atashili,[5] Brian Angus,[6] Trudie Lang[1]

For numbered affiliations see end of article.

**Correspondence to**
Dr Samuel R P Franzen;
sam.franzen@opml.co.uk

## ABSTRACT

**Objectives** In 2013, the WHO stated that unless low-income and middle-income countries (LMICs) become producers of research, health goals would be hard to achieve. Among the capacities required to build a local evidence base, ability to conduct clinical trials is important. There is no evidence-based guidance for the best ways to develop locally led trial capacity. This research aims to identify the barriers and enablers to locally led clinical trial conduct in LMICs and determine strategies for their sustainable development.

**Design** Prospective, multiple case study design consisting of interviews (n=34), focus group discussions (n=13) and process mapping exercises (n=10).

**Setting** Case studies took place in Ethiopia (2011), Cameroon (2012) and Sri Lanka (2013).

**Participants** Local health researchers with previous experiences of clinical trials or stakeholders with an interest in trials were purposively selected through registration searches and snowball sampling (n=100).

**Primary and secondary outcome measures** Discussion notes and transcripts were analysed using thematic coding analysis. Key themes and mechanisms were identified.

**Results** Institutions and individuals were variably successful at conducting trials, but there were strong commonalities in the barriers and enablers across all levels and functions of the research systems. Transferable mechanisms were summarised into the necessary conditions for trial undertaking, which included: awareness of research, motivation, knowledge and technical skills, leadership capabilities, forming collaborations, inclusive trial operations, policy relevance and uptake and macro and institutional strengthening.

**Conclusions** Barriers and enablers to locally led trial undertaking exist at all levels and functions of LMIC research systems. Establishing the necessary conditions to facilitate this research will require multiple, coordinated interventions that seek to resolve them in a systemic manner. The strategies presented in the discussion provide

## Strengths and limitations of this study

► This research represents one of the few empirical studies into the barriers and enablers to locally led clinical trial conduct in low-income and middle-income countries (LMICs) and presents a conceptual framework and strategies for developing sustainable locally led trial capacity.

► Although the broad scope of the research limits the depth of findings, the multicase study design and qualitative methods have successfully captured the key issues influencing locally led trial conduct in diverse contexts.

► Conducting research in only three countries may be considered a weakness. However, this allowed a comparative analysis that could be replicated in other settings, paying attention to the domains outlined in this paper.

► Purposive sampling may have biased the results towards an LMIC researcher viewpoint, but also enabled a focus on the key agents of change. Comparison with wider literature suggests the findings are congruent with international experience.

► This study adds robust evidence to much of the opinion and experience-based framings of research capacity and offers additional empirical insights and novel explanations that warrant further investigation.

an evidence-based framework for a self-sustaining capacity development approach. This represents an important contribution to the literature that will be relevant for research funders, users and producers.

## INTRODUCTION

It is widely accepted that to improve the health and development status of low-income

BMJ

and middle-income countries (LMICs), more research is required into health conditions that cause the greatest burden of disease.[1–4] As much as possible, this research needs to be conducted within LMICs[4 5] in order define the problems that need to be attended to and 'propose culturally apt and cost-effective individual and collective interventions, to investigate their implementation, and to explore the obstacles that prevent recommended strategies from being implemented'.[6]

Several high-profile calls for action have been initiated over the past three decades,[1 3 4] most recently in the 2013 World Health Report that stated that 'all nations should be producers and users of research'.[2] However, despite some progress,[7–9] most research is led by high-income countries,[9] and many LMICs still lack capacity to self-sufficiently undertake research[2] and translate findings into policy.[9] Therefore in most circumstances, gains in health research do not appear sustainable without continued foreign support,[10–12] which is itself questionable in light of recent trends in development assistance.[13 14]

A possible explanation for the lack of progress is that current guidance for capacity development is scarce and too generic to be useful,[15] largely owing to a lack of empirical data on national health research systems[16 17] and development strategies.[18] This situation has led to increasing calls for evidence to guide health research capacity strengthening in LMICs.[19–21] This call for research is particularly pertinent to clinical trials because although they are considered to be vital for generating the necessary evidence to improve health outcomes in LMICs,[2 3] development of self-sufficient trial capacity has proved elusive. Most trials remain foreign led, and they are considered a challenging research design to conduct in LMICs.[2 22] This is in spite of the 2005 WHO statement that the establishment of Africa-owned research centres capable of running their own clinical trials should be an international priority.[23]

A systematic review of the health research capacity development literature[24] reveals little empirical research exploring the implementation of clinical trials in LMICs,

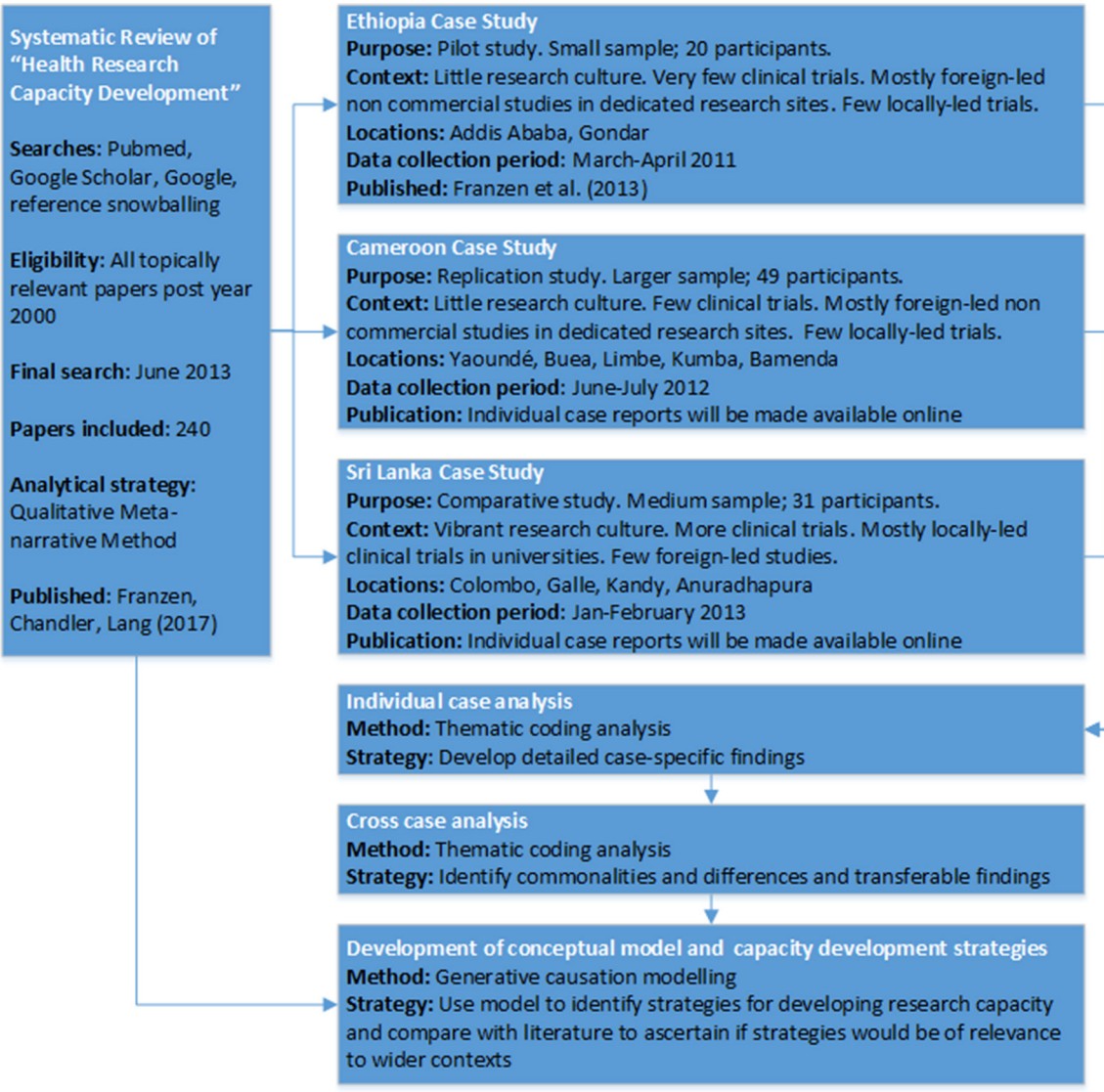

**Figure 1**  Design, settings and sequence of research activities.

and the majority of this is dedicated to developing LMIC capacity to conduct international collaborative trials,[9] rather than self-sufficient capacity to lead their own.[25 26] Indeed, the systematic review only identified three papers in the literature that were dedicated to considering how locally led trial capacity could be developed, and none of these were empirical.[24] As such, development of locally led trial capacity has been largely ignored.[27] This paper aims to contribute towards filling this important evidence gap by identifying the key barriers and enablers to locally led trial conduct in LMICs and developing evidence-based and tailored strategies for sustainable clinical trial capacity development.

## METHODOLOGY

We used a prospective, multiple case study design with qualitative research methods. The design, settings and sequence of research activities are outlined in figure 1. The use of pilot, replication and comparative case studies is suggested where little evidence exists to guide case study design.[28] Accordingly, the first case study in Ethiopia was designed as a pilot to explore issues affecting locally led trial conduct and develop a preliminary conceptual framework. The second, larger case study in Cameroon assessed if the pilot findings and conceptual framework were relevant in a similar context. The final case study in Sri Lanka was conducted to determine if the previous findings and conceptual framework were transferable to a context where locally led trials were more common. A parallel systematic review on health research capacity development was also conducted to determine if case study findings were more widely generalisable. The methods and findings of this review are published as a separate article.[24]

In all case studies, local health researchers with previous experiences of clinical trials or stakeholders with an interest in trials were purposively selected. Potential participants were identified first through trial registration and publication searches, approaching individuals listed on the Global Health Trials website,[29] and subsequently snowball sampling. According to their profile, participants were selected to take part in interviews, focus groups or process mapping exercises. Interviews were used to explore management, governance and other sensitive issues that would not be appropriate for group discussion and when scheduling difficulties meant that group discussions were not possible. Focus groups were conducted with participants who had a variety of research experiences, in order to explore a breadth of perspectives. Process mapping exercises were used with specific research teams who had recently conducted a clinical trial. The purpose of this exercise was to systematically walk through and map the process of conducting a clinical trial.

Research exercises and questions were tailored to the respondents' experience and to explore emerging themes. All exercises were semistructured, conducted in English (by SRPF) and broadly explored the barriers and enablers to trial conduct at all levels of the research system: macro, institutional, individual and operational. Field notes were reviewed shortly after the research exercises to identify emergent themes and determine data saturation. In the Cameroonian and Sri Lankan case studies, sufficient participants were recruited to reach saturation of themes. However, due to the smaller sample size in the Ethiopian pilot study, saturation of themes was not apparent. Repeat interviews were not conducted.

In Cameroon and Sri Lanka, written informed consent was obtained from all participants, and research exercises were recorded and transcribed verbatim. In Ethiopia, participants said that they would be more comfortable giving verbal informed consent and not being audio recorded. Accordingly, detailed notes were taken with quotes noted as near verbatim as possible, detailing identification numbers. Of the participants approached, none refused to take part. Ethics approvals were obtained from

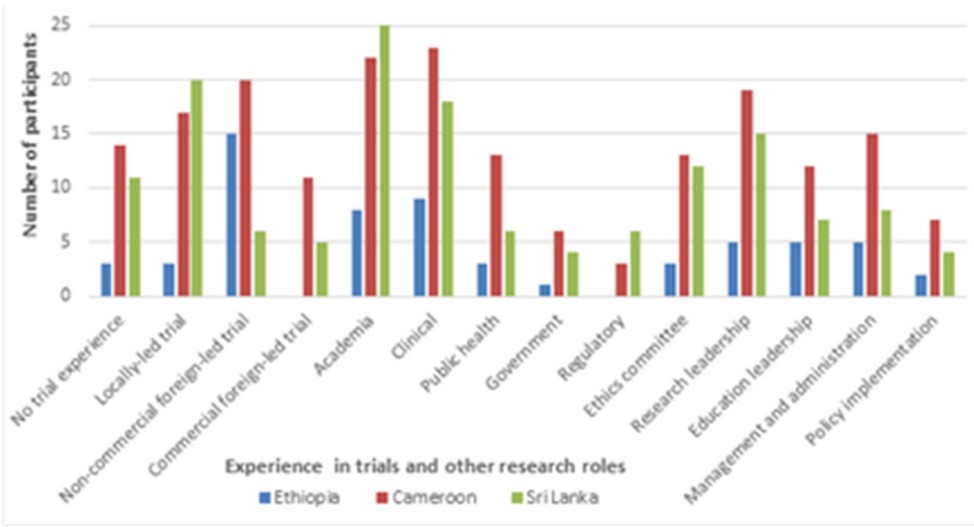

**Figure 2** Clinical trial and other research roles held by participants in the three case studies.

**Table 1** Number and type of research exercises by case study country

| Research exercise | Total | Number of research exercises by case study country | | |
| --- | --- | --- | --- | --- |
| | | Ethiopia | Cameroon | Sri Lanka |
| Interview | 34 | 6 | 16 | 12 |
| Focus group discussion | 13 | 3 | 6 | 4 |
| Process mapping | 10 | 1 | 6 | 3 |
| Total | 57 | 10 | 28 | 19 |

The University of Oxford, UK; The University of Buea, Cameroon; The University of Yaoundé, Cameroon; The National Ethics Committee, Cameroon; and The University of Sri Jayewardenepura, Sri Lanka.

Each case study was first analysed and reported as a separate standalone case, after which cross-case analysis was conducted. Transcripts were analysed by thematic coding analysis[30] using Nvivo qualitative data analysis package (QSR International Pty, V.9, 2011). Data were coded inductively, and conceptual models were developed by drawing on generative causation approaches used in realist research.[31] These approaches help to identify context–mechanism–outcome configurations that can explain when and how elements of the system interact with one another to produce a given outcome.[32] They are therefore useful for identifying and developing strategic recommendations. Identification of these configurations was facilitated through the use of the relationship modelling features of Nvivo. SF completed coding with consultation and agreement from other authors (TL, CC and BA). Findings were reviewed and commented on by all authors.

Findings of the Ethiopian case study,[33] literature review,[24] research protocol and detailed methodology and individual case reports are available online.[29]

## RESULTS
### Study population and research context
One hundred participants were recruited: 20 in Ethiopia, 49 in Cameroon and 31 in Sri Lanka. The clinical trial and other research roles held by the participants are shown in figure 2. Participants usually had several jobs that could cover multiple research roles.

In all case study countries, most research was preclinical, using descriptive designs such as case studies or cross-sectional analysis. Of the experimental research done, only a small proportion were clinical trials. The clinical trials conducted by participants mostly investigated the use of previously approved therapeutics to improve the treatment of locally important diseases, with a few investigating operational topics including behavioural interventions (for instance, to improve

drug adherence). Investigation of novel therapeutics was rare, although in Sri Lanka the use of clinical trials to determine the effectiveness of traditional medicines was common.

Among the participants, a total of 34 interviews, 13 focus group discussions and 10 process mapping exercises were conducted. A breakdown of the number of research exercises by case study is shown in table 1.

### Barriers and enablers to locally led trial conduct
This article compares and synthesises the key findings from three case studies to identify transferable strategies for developing locally led trial capacity in LMICs. The complete list and description of the barriers and enablers to trial conduct that were identified in the case studies is shown in online supplementary file 1. This table organises the findings by the functions of the research system[34] and compares across case studies to examine differences. Findings from the systematic review published as a separate article[24] are also compared for reference later in the discussion.

Although some barriers and enablers were more or less influential in different research contexts, the majority were present in every case study. There were no contradictory findings whereby specific barriers and enablers were not considered important. A conceptual model of the necessary conditions for locally led trial conduct was developed to demonstrate the interaction between these common barriers and enablers (figure 3). The elements of this model, justification for the mechanisms and example respondent quotations are presented below. It is important to note that the conceptual model takes an enabling perspective to identify strategies for developing locally led trial capacity. However, enabling mechanisms were often not present within the case study contexts but rather represented respondent views on what would help resolve barriers and facilitate trial conduct.

### Individual level
#### Awareness of health research and clinical trials
There was wide agreement between participants in all case studies that greater awareness of the benefits of clinical trials and health research was needed to foster more proresearch cultures and locally led trials. The concept of awareness, as described by participants, encompassed two main aspects.

First, an overall understanding of the concept of modern biomedical research was thought to be important for developing a future cadre of health researchers with a positive and interested attitude towards clinical trials and health research more generally. Second, in many case study institutions, especially healthcare, practitioners and decision makers often did not see that evidence-based medicine could improve patient care or were resistant to it on the grounds it could limit their autonomy in treating patients. Overcoming this resistance was reported by local researchers to be critical for ensuring a more positive

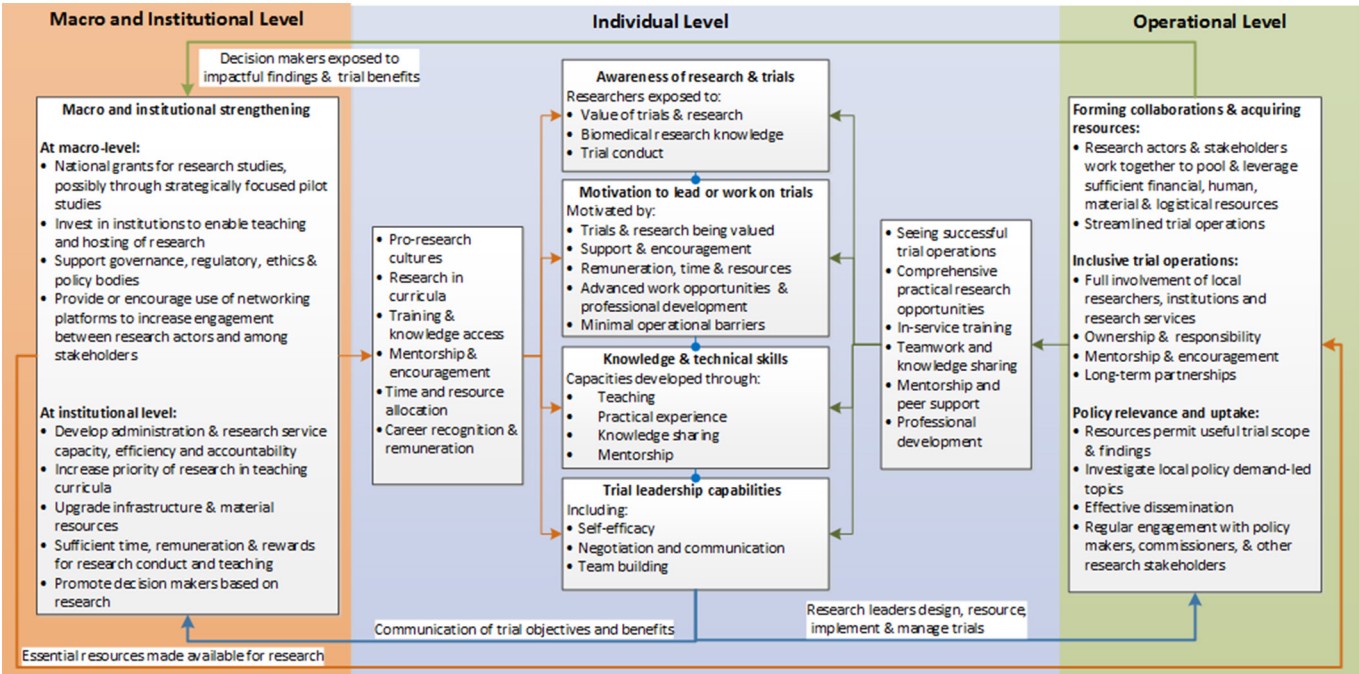

**Figure 3** Conceptual model of the necessary conditions for locally led trial conduct.

research culture and securing the allocation of resources to allow more research. Accordingly, awareness activities that could convince individuals of the legitimacy of evidence-based medicine for improving population health and the value of clinical trials for contributing to the evidence base were reportedly needed. Exposure methods suggested by current researchers and clinical trial practitioners included: increasing research and clinical trial modules in university curricula, mentorship, knowledge sharing events such as seminars and workshops, training courses and access to knowledge resources and opportunities to work on trials. In all case studies, seeing trials conducted within individuals' own institutions was seen as particularly important for enhancing this awareness.

> People need to be made aware of these trials. I think there has to be more explanation of the concept of clinical trials in Sri Lanka. It's not in our normal day-to-day priorities you know, it's not in our work ethic, the value of clinical trials and the application of findings locally. This attitudinal change can be brought about by increased awareness, having open forums and incentives… We have to show the outcome of these trials. That will show that they are important. Then people might end up doing some! (Sri Lankan academic clinician and trial investigator)

### Motivation to lead or work on clinical trials

Personal motivation for research leaders and staff to conduct clinical trials and health research was a very important theme in all case studies. This was because for most individuals, research was a discretionary activity that they can choose to undertake, usually alongside many

other competing priorities. If suitable incentives were not present, individuals were unlikely to undertake trials or may choose to work in external national or international institutions that provided better incentives, resulting in brain drain from local institutions.

Within dedicated research sites in all case study countries and Sri Lankan academic institutions, potential researchers willingly conducted trials because research was required for career progression and supported through providing time, incentives and resources for research. However, within academic institutions in Cameroon and Ethiopia and healthcare institutions in all case countries, few research incentives were provided. Even when research was linked to career progression, it often did not lead to comparatively better working conditions. As such, research was often seen as a side activity that needed to compete with private practice and other duties but frequently failed to do so because of relatively poor incentives.

> Research is restricted to academic individuals. Doctors who are not academic do not get any benefit in money or recognition and career development for doing research. Research is not appreciated as part of career development by the Ministry of Health. (Sri Lankan academic and trial investigator)

Perceptions that trial operations would be difficult and time consuming, inadequate resources, negative attitudes towards research and lack of peer support further decreased motivation to conduct trials. To encourage research, academic and healthcare staff felt that institutions needed to provide allocated time for research,

 

financial incentives and link research to career progression and better working conditions.

> The tendency is that they give you the fellowship but after a year it is done, then it's like 'You're on your own.'…Now the person comes back but he has no means, no ability or opportunity to implement anything he has learned. So consequently it would appear to him as a complete waste of time. (Cameroonian academic)

However, some employees in all case countries still chose to conduct trials despite few incentives. The motivation for these 'unconventional' investigators was driven by the desire for personal and professional development, opportunities for responsibility, challenging work and international or peer recognition. Therefore, these career and personal growth incentives were sometimes sufficient to offset lack of other incentives, at least for a time.

> I felt like I was recognised as a scientist when they allocated the funds for me to manage. They recognised that I could be a leader and they have given more responsibilities', and that gave me more courage. It also motivated me in the sense that I would always be the principal investigator, so if, for example, there is a presentation somewhere I will probably be able to go for this presentation and also stand up among other peers or scientists, among everywhere, and talk. (Cameroonian clinical trial investigator)

### Knowledge and technical skills to undertake trials

There was wide agreement in all case studies that more staff with the knowledge and skills to lead and work on trials were needed. Indeed, participants from Ethiopia and Sri Lanka argued that lack of suitably skilled trial staff was one of the greatest barriers to their conduct.

All case studies were in agreement that the lack of skilled staff was driven by limited attention to research methods in undergraduate curricula and continuing education, especially in healthcare fields. However, capacity to teach research, especially clinical trials, was also limited. Access to knowledge resources was seen as a possible substitute to enable interested individuals to pursue independent learning. For ease of access, internet-based open-access journals and e-learning were preferred, and HINARI[35] was widely cited as an extremely useful resource. However, many participants reported limited availability of these knowledge resources and said that regular training based on local research conditions was still required.

> The clinicians, they are not research oriented. I think if there can be improvement in teaching of research methodology in the curriculum of medical schools that would help. If there can be continuous medical education sessions, or refresher courses on research methodology and the importance of carrying out research, it would go a long way to improve upon

the knowledge and the technical knowhow of the personnel, and facilitate the necessary research enormously. (Cameroonian clinician and trial coinvestigator)

Although developing faculty teaching capacity and providing improved access to learning resources was important, participants in all case studies considered practical trial experiences to be essential for developing technical skills. In Cameroon and Ethiopia, lack of these practical learning opportunities was considered to be one of the main barriers to the development of human resources for trials.

> Getting exposed to different aspects of research and working with different groups of people is an experience you really can only have if you are part of it [clinical trial]. Your knowledge increases, your understanding, you have to think deeper. Interacting with high profile professors who are very experienced, I learned a lot. I was improving so by the time I did it the second and third trial, because you've been involved in all of this, you can stand and talk very broadly. (Cameroonian trial project coordinator)

### Trial leadership capabilities

In all case studies, it was clear that undertaking successful trials required technical knowledge and skills and specific leadership capabilities, namely: self-efficacy, negotiation and communication skills and team building.

Self-efficacy (often described by participants as confidence or belief that they could successfully conduct a trial) was considered to be very important for trial leadership in all case studies. This is because it reportedly gave investigators the belief that they could lead trials in challenging environments and the ability to react positively and persist in the face of common operational barriers.

> I have never been involved in any other trials but he has [refers to trial experienced senior colleague PM.4.PPT.2], and I think that was what gave us the strength to strike out on our own and figure out yes, we could do this! I'm basically a parasitologist, I'm a lab person, but the professors' input on the trial really helped. (Sri Lankan head of academic department)

Negotiating and communication were considered particularly enabling to research leadership in the Sri Lankan and Cameroon case studies because these skills could reportedly help forge collaborations and bring all the necessary stakeholders together to work towards common goals, including securing institutional buy-in and investment. Often this was achieved through particular communication strategies that could encourage individuals and institutions to support, not hinder, trials. Team building skills were important for making trial operations more efficient by developing effective team-working environments.

Although it was not exactly clear how these leadership capabilities were developed, they were often associated with positive trial work experiences that provided opportunities for: involvement in the whole research process; responsibility and challenging work; exposure to research role models; and working environments that encouraged contributions, peer support and taking initiative. However, such environments were rare in Cameroon and Ethiopia and healthcare institutions in Sri Lanka.

> The [PI] has a career development mentality, so by the time you are coming out [finishing the trial] you are totally different from the way you were before…When someone is only given instructions I bet you will not learn anything. In our group you are taught everything but it's not like the professor does everything, everyone is involved, if you are leading an aspect you do it right up to the end, the professor guides you, but you have to show him what you have at the end. If he's not available to go for a meeting another team member will go, so that encourages you like 'oh he must trust me up to a level where he lets me represent him and present our study'. (Cameroonian research assistant)

### Operational level

The in-country conduct of clinical trials was viewed by respondents in all case countries as very important because such trials were thought to provide locally relevant and high-quality data with which to fill evidence gaps and tailor international guidelines. However, trial conduct was also seen as critical for developing institutional research capacity. Indeed in Cameroon, some current researchers considered this institutional impact to be as important as evidence outputs, and in Sri Lanka, clinical trials were actively encouraged as a capacity development, rather than purely health development tool.

> They think they can attract foreign revenue here. That's the treasury side. Also the other thing is that at the moment we don't have the ability to conduct big research here, that's the funding and facilities we don't have, so it is better to have some international research.… then if we initiate the international multicentre trials here at least, then one day, through capacity building, we can do our own thing better than today. (Sri Lankan regulatory board member)

However, the ability of trials to achieve these beneficial outcomes was variable and dependent on how they were managed and led. To support evidence and capacity development impacts, three elements of trial operations seemed important: collaboration: inclusive trial operations, and policy relevance and uptake.

### Forming collaborations and acquiring resources

The importance of collaboration for enabling locally led trials was reported by many trial teams in every case country. International collaboration was very helpful for enabling research that was beyond local capacity constraints. In Cameroon and Ethiopia where local resources and funding were minimal, collaboration with foreign groups was near essential. This was because foreign collaborations provided finances, access to material resources and human expertise, logistical and administrative support and credibility and support with grant application. Indeed in all case studies, successfully gaining international funding was almost always associated with foreign collaboration or assistance. Although some trial teams were successful in forming international collaborations, respondents from all case studies commented that this was difficult due to a lack of networking opportunities and contacts, insufficient institutional capacity to attract collaborators or local research topics being of little international interest.

> Participating [in X consortium] has given us this opportunity to build collaborations with very good researchers. People now know that we exist, and that is good. We have the capacity now to go and develop. All my students are going to learn clinical training. I don't have any problem with that now I have an infrastructure. The platform where they can do good research has automatically enhanced the quality of training. (Cameroonian head of research department)

Local collaboration was also very enabling when it was achieved because it could bring disparate local resources together to reach a self-sufficient critical mass. Collaborations that went beyond research-producers were cited as particularly helpful; for instance, working with hospitals, schools and ministries permitted pooling and sharing of resources such as staff, transport and laboratory facilities. However, forming local collaborations was reportedly rare due to poor local networking, competitive or negative research cultures and preference for international partners.

To facilitate local and international collaborations, respondents stated that better networking was needed. This could reportedly be achieved through developing national researcher registries, holding networking events and providing access to online research networks. However, to make collaboration more appealing for partners, better institutional capacity and research support systems were reportedly required. Indeed, in Cameroon, several respondents stated that potential partners were reticent about collaborating due to the level of investment that would be required to conduct clinical trials.

> Networking that's a big gap. You see we need awareness of each other first. In Cameroon, there's smart people but the knowledge just stays there, nobody uses it. In Africa, people don't know each other exist and so cannot maximise resources and cannot work together. We need to map out expertise on a system or database. It will also give an opportunity for North-South collaboration. (Cameroonian academic)

## Inclusive trial operations

Experience in trial work was critical to the development of knowledge, technical skills and leadership capabilities to undertake clinical trials. It could also provide a platform for promoting awareness and positive attitude to trials. Material and financial resources provided through clinical trials was also often instrumental in developing institutional capacity to undertake subsequent trials. However, for these outcomes to be successfully achieved, it was clear that trials needed to be managed with capacity development in mind.

First, it was important that trials were conducted within local institutions and gave potential researchers and decision makers the opportunity to understand what conducting a trial involves. Second, trials needed to use as many local staff as possible, involve them in all processes and provide opportunities for responsibility and challenging work so that technical and leadership skills and motivation could be developed. Third, material and financial resources needed to be routed through local institutions so that they could retain trial resources and to develop administrative expertise in providing research services.

Locally led trials were generally considered the best model for achieving these capacity development ideals because in the majority of cases they were conducted within local institutions, all trial staff were locally sourced and there were more opportunities for full involvement, responsibility and ownership of the trial. Furthermore, all material resources and finances arising from the trial were usually managed and retained by the local institution. However, locally led trials reportedly had limited ability to develop capacity in more advanced skills because financial, human and material resources were often lacking. Poor administrative services and bureaucratic procedures also encouraged local investigators to set up parallel structures or route their research through foreign institutions, thereby reducing opportunities for capacity development.

As presented above, long-term foreign collaborations were also reported to provide excellent capacity development opportunities. However, on most short-term trial collaborations, and even one long-term partnership, this level of local inclusion did not occur. This was because local staff were frequently only given support roles, and they were not involved in planning, analysis and write-up stages. Material capacity development was variable, and sample analysis was often done abroad, so laboratory capacity was not always developed. Therefore, although short-term collaborations could provide useful junior trial experiences and some material gain, self-sufficient capacity was not often developed.

> On the other [foreign-led short-term] collaborations, they just wanted us to collect the data. So you see we didn't learn and develop…But on our [X trial - locally-led] we got to really face a lot of challenges and overcame them and then through that we

developed. I think one proof that the [X trial] was very instrumental in building our capacity was that we've been able to develop some more ideas in a more refined manner. We have a saying that 'the son shows maturity when he picks up his arrow and goes hunting'. It's an African saying. You know that the son is mature when he picks up his arrow. He doesn't wait for his father. He doesn't wait for his uncle. He just goes hunting. This is what I think I have been able to do more with the other [locally-led] trials. (Cameroonian clinical trial investigator)

## Policy relevance and uptake

Most participants in all case studies considered locally led trial evidence to be more useful for policy than foreign-led studies' evidence because local investigators would be more likely to investigate policy-relevant topics and have the best relationships with policy makers. However, the ability of local trial evidence to actually influence policy was often prevented by: research outputs being piecemeal and of limited scope, poor relationships between research producers and users and policy makers lacking capacity or interest to demand or use research.

> Currently we just do ad hoc research, you know, whatever takes our fancy. Most research is not useful and done individually so it is fragmented so we can't make recommendations based on these individual studies. We need a coordinated and strategic approach but there are no priority areas. The Ministry of Health should be doing this but they don't. (Sri Lankan Academic)

In contrast, decision makers stated that foreign-led studies could sometimes be better than locally led trials at influencing policy. This was due to their research outputs often being of greater scope and quality and having more credibility and resources to dedicate towards disseminating research and influencing policy. Furthermore, research topics were often locally relevant, especially where strong local leadership was present. However, such foreign dedication to policy impact, while desired, was only rarely reported in Cameroon and Sri Lanka, and not in Ethiopia, and then only done by long-term partnerships. Indeed, one common criticism of short-term collaborations was that they often failed to involve local stakeholders in research planning and did not disseminate findings locally.

To facilitate research uptake, both research producers and users were in agreement that local research needed greater investment to ensure research had sufficient scope to be meaningful, and there needed to be earlier and more frequent engagement with policy makers to ensure policy relevant investigation and dissemination.

> It's true that it's a problem trying to get along with the authorities, but once you get to understand them and they know the value of your work then it becomes easier to translate, to advocate for these interventions

that are life-saving. It's easier to integrate with the policymakers if they know you and you come to them pretty often. (Cameroonian trial coordinator)

## Macro and institutional level

In Ethiopia and Cameroon, macro and institutional level deficiencies meant that only a few exceptional individuals were able to conduct trials within local capacity constraints, and this was rarely sustained. Furthermore, in all case countries, limited resources and operational barriers reduced motivation to conduct trials, prejudiced grant applications and international collaboration opportunities, led to bypassing local institutions and limited the usefulness and capacity development potential of trial research.

An increase in government investment for local research in Ethiopia and Cameroon was considered essential because most system inadequacies were ultimately attributed to lack of financing. Although international grants and clinical trials could provide financial and material resources, their provision was always limited to donors' thematic focus and made local researchers dependent on foreign collaboration. Therefore, to enable self-sufficient research, more local investment was reportedly required.

We have a freezer full of important samples that need to be analysed, but we have no specific funding or resources for that. So they just stay in the freezer. We also need guidance on how to do this. (Ethiopian researcher working on a foreign-led trial)

Participants emphasised that such investment could be in the form of small-scale pilot grants designed to stimulate and strengthen local research, and indeed the positive effects of such grants was felt by participants in the Sri Lankan case study. However, research producers and users were in agreement that local grants needed to be more demand-driven and strategically provided otherwise research outputs would continue to be fragmented and have limited usefulness for policy.

We need to develop and support a research culture. We need grants for beginner researchers to do research and get practice - this would take away the phobia. When the phobia has gone there will be floods of research. We need to open our eyes and see what can be done…Even small research will be an eye opener and the phobia will be gone. (Ethiopian junior academic)

In all case studies, regulatory and ethical bodies lacked sufficient capacity to govern research. In Sri Lanka, this was considered a key bottleneck to further expansion of clinical trials. Efforts to develop governance capacity were present in all countries, but these were largely driven by interested individuals or poorly resourced government departments, and most regulatory procedures lacked legal backing. Administration was seriously problematic in all case countries due to overly centralised, bureaucratic and hierarchical structures that were often resistant to research. To resolve these issues, participants suggested that regulatory and ethical review boards needed greater investment and capacity building, procedures should be streamlined and there needed to be greater accountability put on bureaucrats, including meritocratic promotion based on research experience. Administrators also argued that research services required a greater proportion of research overheads and more inclusion in grant application and management processes if they were to improve and support researchers.

There are a lot of complications, a lot of administrative bother. You get into a process where, 'Oh, you have to see this person, you need to see this other person, you need to go and see this person. You get this before you see this other person who will now give you authorisation to see this other person.' Basically the procedures are very complex. (Cameroonian academic researcher)

Participants stated that research leaders had an important role to play in driving these changes by advocating the importance of clinical trials for health outcomes and institutional capacity. However, to make these arguments plausible, decision makers stated that local researchers needed to demonstrate these benefits through influencing policy and developing local research capacity.

You cannot see a building from the state that is a research building. It is not because the state does not have money for that. Those who are making decisions on behalf of the state have a lack of interest for research. It needs pressure from the deans to ensure the government allocates money and the money goes to the right equipment. But the leaders are not leading. (Cameroonian academic)

## DISCUSSION
### Summary

This paper has described the key barriers and enablers influencing locally led trial conduct within three case studies in Ethiopia, Cameroon and Sri Lanka. Although different country research systems and institutions and individuals within them were variably successful at conducting trials, there were strong commonalities in the underlying determinants across all levels and functions of the research system. These transferable mechanisms were summarised into a conceptual model of the necessary conditions for locally led trial undertaking. The model draws together the often fragmented and individually addressed issues facing clinical trial conduct in LMICs into a research systems perspective.[34]

A detailed comparison of the barriers and enablers identified in the three case is presented in online supplementary file 1. This comparison and the conceptual

model (figure 3) suggests that Sri Lanka was more productive in terms of its clinical trial and research outputs compared with Ethiopia and Cameroon due to an enabling research environment that can be traced back to proresearch cultures at multiple levels. Resources for research were more available within Sri Lanka in terms of national grants, better quality infrastructure and equipment and stronger incentives for conducting research, at least within academia. These basic prerequisites supported locally led trials and meant that local researchers were not dependent on international collaboration or parallel research structures. The resulting higher volume of national research is, in turn, likely to explain why networking, exposure to clinical trials and self-efficacy were less problematic in Sri Lanka. Finally, the availability of locally generated evidence appeared to meet most of the needs of policy makers because a preference for international evidence was not mentioned. This may explain the greater buy-in for research at policy level, which is evidence by greater research investments.

However, while this was true for academia, the problems facing research in healthcare environments in Sri Lanka were similar to those in Ethiopia and Cameroon. Stewardship and governance capacity, and the availability of human resources capable of conducting clinical trials, were also limiting factors in all countries. This suggests that while specific national investments can be helpful, a whole-of-systems approach is needed to comprehensively address the issues facing locally led research in developing countries.

### Strategies for developing sustainable health research capacity in LMICs

The congruence between the barriers and enablers identified in the three case studies with the health research capacity strengthening literarture[24] (presented in online supplementary file 1) suggests that the conceptual model is likely to be relevant to other LMIC research contexts and possibly other types of health research beyond clinical trials. Given this potential for generalisability, we adapted the conceptual framework into long-term and self-sustaining strategies for increasing locally led trial conduct in LMICs.

As presented in table 2, we divided our strategies into four goals: (1) fostering proresearch cultures, (2) developing trial leaders and staff, (3) providing a facilitative operational environment and (4) ensuring trial research has an impact. These goals, and the logic by which they can promote locally led trial conduct, were identified by grouping the lower level theory that was empirically developed in the conceptual framework into categories of higher level mechanisms that may ultimately lead to the desired outcome. To ensure the strategies are specific, action orientated and context sensitive, each includes an implementation plan, mechanism of change, agent responsible and context where the mechanisms are likely to be most important.

Strategies under 'Fostering pro-research cultures' focus on generating top-level buy-in to secure investment, generate support and appreciation for trial research and increasing the pool of potential researchers willing and confident enough to conduct trials. 'Developing trial leaders and staff' concentrates on resolving skills gaps of academics and healthcare staff so that they can undertake trials and on developing future research leaders that have the capabilities to successfully manage trials in challenging environments, support the development of local staff and institutions and can act as champions for change. 'Providing a facilitative operational environment for trials' aims to reduce operational barriers to trial conduct and increase material resources so that future trials can be conducted with greater scope, quality and ease, therefore making trial conduct within local institutions a more attractive option. 'Ensuring trial research has an impact' aims to make clinical trial evidence useful for policy and to demonstrate that local research is credible, valuable and offers a good return on investment so that proresearch cultures and support for trials is reinforced.

### Strengths and limitations

This research represents one of the few empirical studies into locally led clinical trial undertaking in LMICs. We hope this will encourage further research in this area, potentially through adapting and applying our methodology in other contexts. The phased, multicase study approach has successfully captured the key issues influencing locally led clinical trial conduct in diverse contexts. Similarity with the parallel systematic review findings[24] indicated sufficient transferability to develop a common conceptual model and recommendations for developing locally led trial capacity that will be relevant to many LMIC research contexts and potentially other types of health research.

While the strategies presented in this paper are aligned with established guides for health research capacity development,[8 15 36 37] to our knowledge they are the only set of recommendations that are explicitly empirically based, follow a conceptual framework and provide sufficient detail to determine suitability for specific contexts. Since the paucity of empirically grounded, contextually relevant and conceptually informed guidance for health research capacity development is a recognised problem,[15 16 18] this study represents an important contribution to the literature and goes some way to contribute to the evidence called for in the 2013 World Health Report.[2]

Although individual capacity development has long been considered important,[9] empirical demonstration of the latent factors influencing clinical trial decision making and the central importance of research leaders in conducting trials in developing capacity and championing change is largely novel. Furthermore, while good practice in health research capacity development is a frequent point of debate,[12 38] determining how best to

**Table 2** Recommendations to develop sustainable locally led trial capacity in LMICs

| Goal | Logic for change | Strategy | Implementation plan | Mechanism of change | Agent of change | Contextual relevance |
|---|---|---|---|---|---|---|
| Foster proresearch cultures | Encourages top-level investment and prioritisation of trials Encourages institutional staff and decision makers to support not hinder trials | Explain trial and research methods and potential benefits for patients, institutions and individuals | ▲ Research and trial exposure in education and workplaces<br>▲ Engage and inspire through mentorship<br>▲ Access to training and knowledge resources<br>▲ Organise seminars and workshops | Increases awareness and desire to conduct trials and top-level buy-in and support for trials | ▲ Institutional level<br>▲ Research leaders<br>▲ International actors | Where negative research cultures or lack of interest in trials impedes operations and prevents investment |
| | Increases pool of researchers willing and confident enough to conduct trials and reduces brain-drain | Provide opportunities for institutional staff to see trials conducted and practically get involved | ▲ Conduct trials in institutions and involve local staff<br>▲ Allow wider participation through exchange placements<br>▲ Seeing successful locally led trials most encouraging | Increases awareness and desire to conduct trials. Increases motivation and self-efficacy by reducing perception that trials are difficult | ▲ Research leaders | Where skilled or junior staff show little inclination towards trial undertaking Where brain-drain problematic |
| | | Provide intrinsic and extrinsic incentives for employees to conduct or get involved in trials | ▲ Financial rewards and salaried time for research<br>▲ Research linked to career progression leading to better working conditions<br>▲ Provide rewards, appreciation and applauding research | Increases motivation to conduct trials | ▲ Macro and institutional level<br>▲ Research leaders | |
| | | Provide facilitative operational environment for trials | See section below | Increases motivation and self-efficacy to conduct trials by making trials more achievable | See section below | |
| Develop trial leaders and staff | Human resources for research are essential for increasing trial conduct, either locally or foreign led Resolving key skills gaps is needed for researchers to gain funding and conduct trials | Provide basic and advanced skills training. Focus on clinical trials and key skills gaps. Ensure regularity and sustainability. Best if locally applicable. | ▲ Increase research components in educational curricula<br>▲ Provide continuing education in workplaces<br>▲ Skills courses and workshops<br>▲ e-Learning and distance learning<br>▲ Fellowships and advanced degrees<br>▲ Use train-the-trainer models<br>▲ Use more applied teaching techniques | Improves knowledge, develops technical skills, reinforces motivation and increase self-efficacy | ▲ Macro and institutional level<br>▲ Research leaders<br>▲ International actors | Where extant expertise is insufficient to meet demand Where staff have key skills gaps that prevent or impede trials Where there are insufficient research leaders Where research leaders lack leadership capabilities |
| | Research leaders needed to conduct trials, foster proresearch cultures, provide training and mentorship, develop new research leaders and advocate for greater investment | Provide practical research experiences on trials. Locally-led trials and long-term foreign partnerships usually best. | ▲ Provide facilitative environment to encourage complete conduct of trials in institutions (see section below)<br>▲ Offer full involvement, responsibility and challenging work to local staff<br>▲ Provide mentorship and comprehensive training | Most effective technique for mastering technical skills and developing leadership capabilities. Increases motivation. | ▲ Research leaders<br>▲ Foreign collaborators | |
| | | Provide knowledge sharing and mentorship opportunities | ▲ Organise seminars and workshops<br>▲ Encourage teamwork and on-the-job knowledge sharing by developing leadership capabilities<br>▲ Coordinate mentoring relationships<br>▲ Use international networks if unavailable locally | Shares knowledge and provides support that increases knowledge, technical skills, motivation and self-efficacy | ▲ Institutional level<br>▲ Research leaders<br>▲ Colleagues<br>▲ International actors | |
| | | Provide open, easy access to knowledge resources | ▲ Provide libraries, computers and reliable internet<br>▲ Ensure access to HINARI and open-access journals<br>▲ Supply e-learning and offline research guidance | Supports independent learning that increases knowledge and motivation | ▲ Macro and institutional level<br>▲ International actors | |

**Table 2** Continued

| Goal | Logic for change | Strategy | Implementation plan | Mechanism of change | Agent of change | Contextual relevance |
|---|---|---|---|---|---|---|
| **Provide facilitative operational environment** | Reduces barriers to trial conduct that increases self-efficacy and motivation to undertake trials | Provide funding for clinical trials that is sufficient to allow research of useful scope | ▲ Offer international grants exclusively for LMIC researchers<br>▲ National pilot grants for early researchers to gain experience and build portfolios so they can compete for international funding | Even modest grants can enable simple but important locally led trials.<br>Improves chances of gaining more competitive funding. | ▲ Macro level<br>▲ International actors | Where trials are prevented due to operational barriers or material resource constraints |
| | Makes collaboration more attractive<br>Encourages local and foreign-led research to be conducted through local institutions | Improve research governance and administration procedures and increase capacity to support research | ▲ Promote decision makers based on research experience<br>▲ Streamline procedures, update regulations and introduce greater accountability<br>▲ Early engagement between administrators and researchers<br>▲ Budget research services into grants | Speeds up trial operations and frees investigator's time | ▲ Institutional level<br>▲ Research leaders | Where operational barriers or material resources reduce the quality and scope of trials |
| | Facilitates trials of greater scope and quality and increases capacity development benefits that supports advocacy for greater investments | Strengthen regulatory and ethical review capacity and procedures | ▲ Provide funding and training for review boards<br>▲ Ethics training for investigators<br>▲ Build monitoring capacity, develop legal framework and government backing for regulatory bodies | Ensures trials are safe and ethical, allows more ethically complex trials, speeds up trial operations | ▲ Macro and institutional level<br>▲ Research leaders | Where operational barriers or material resources prevent beneficial collaborations or capacity development |
| | | Develop material resources and infrastructure | ▲ Provide sufficient building space with reliable services<br>▲ Provide advanced and basic laboratory equipment and supplies/maintenance<br>▲ Provide sufficient ICT access with reliable internet | Facilitates operations and enables trials with greater scope and quality | ▲ Macro and institutional level | |
| | | Support local collaborations among research producers and stakeholders, and encourage team working | ▲ Develop networking platforms to identify and bring together all local stakeholders<br>▲ Develop and use research leader skills to improve communication and team working | Leverages resources to reach a critical mass capable of self-sufficiently undertaking trials. Improves trial operations. | ▲ Macro and institutional level<br>▲ Research leaders | |
| | | Encourage valuable foreign partnerships. Long-term partnerships most useful | ▲ Provide international networking platforms<br>▲ Ensure foreign collaborations have sufficient capacity to work within local institutions, without major investment<br>▲ Negotiate partnerships that have strong local leadership, are dedicated to capacity development and ideally conduct trials in local institutions | Enables more resource-intensive research and helps develop local capacities | ▲ Macro level<br>▲ Research leaders<br>▲ International actors<br>▲ Foreign collaborators | |

**Table 2** Continued

| Goal | Logic for change | Strategy | Implementation plan | Mechanism of change | Agent of change | Contextual relevance |
|---|---|---|---|---|---|---|
| **Ensure research is useful and has an impact** | Trials must influence policy and have an impact on health outcomes for them to be considered valuable. Useful and impactful trials develop and reinforce proresearch attitudes by showing benefits and returns on investments | Develop and implement clear research strategy to focus investments around research priorities | ▲ Develop and disseminate clear research strategy<br>▲ Focus local grant funding on key areas and make grants demand led<br>▲ Focus institutional investments on local departments and resources required to meet research goals | Ensures most efficient use of resources and builds an evidence base capable of informing policy changes | ▲ Macro level | Where trial evidence has limited use for policy or is not effectively disseminated |
| | | Develop policy makers' interest and capacity to demand and use research and implement policies | ▲ Foster proresearch cultures and attitudes (see section above)<br>▲ Provide training for policy makers to demand and use research<br>▲ Ensure resources available for policy implementation | Ensures research has an impact and improves patient care | ▲ Macro level | Where research users lack capacity to translate research and implement policies |
| | Increases credibility of locally led trials, which is needed for research leaders to advocate for further investment | Develop research producers interest and capacity to respond to research strategy, produce useful outputs and disseminate findings effectively | ▲ Provide a facilitative operational environment conducive to useful research (see section above)<br>▲ Develop research leaders who can effectively interact with these bodies (see section above)<br>▲ Provide training on research dissemination for publication and policy<br>▲ Ensure time and resources available for disseminating findings | Ensures research findings will be useful for policy and are effectively disseminated to influence policy | ▲ Macro and institutional level<br>▲ Research leaders | Where poor communication and engagement impedes translation of evidence into policy |
| | | Increase engagement between strategists, producers and users of research | ▲ Develop networking platforms to facilitate interaction between these stakeholders<br>▲ Engage early and regularly<br>▲ Dedicated liaisons may be helpful | Builds communication and trust between knowledge cycle actors that facilitate translation of research | ▲ Macro level<br>▲ Research leaders | |

LMICs, low-income and middle-income countries. ICT, Information Communication Technology.

conduct a clinical trial with capacity development in mind has rarely been defined and evidenced.[39] This rhetorical rather than actionable approach towards health research capacity development was a key finding in our previously published literature review,[24] which concluded that sustainable capacity development required dedicated efforts. The findings of this study help to refine and evidence what these dedicated efforts should involve.

Considering a research system as a single case may be disputed by some researchers. This is because traditional cases have distinct boundaries that are investigated in detail.[28] Therefore, the cases presented could be argued to be rather shallow. Furthermore, the lack of inclusion of international stakeholders as participants restricts the perspectives represented in this study. However, the objectives of this research were to try to establish the most commonly encountered 'high order' barriers within research systems that need to be addressed to facilitate locally led trials. Therefore, it was necessary to sacrifice some detail in order to capture broad experiences from the various institutions that make up national research systems. This is a pragmatic approach, but one that D'Souza and Sadana[18] say is needed to know where to focus the limited resources available. Reaching data saturation within the Cameroon and Sri Lanka case studies also helped to ensure that the majority of key issues were identified and comparison with the literature reveals the findings to be aligned with international perspectives. Nevertheless, it would be desirable to validate and triangulate this study's findings across a larger and more diverse sample, possibly using quantitative survey methods that could statistically assess associations between key variables.

It is possible that due to the delay in publication of this article, the situation may have changed within the case-study countries. Indeed, where efforts were being made, the trajectory would predict progress in clinical trial capacity. Nevertheless, improvement in research systems has historically been slow,[24] and the findings are therefore likely to remain valid for many LMICs. This is supported by recent contributions to the literature from WHO-TDR and ESSENCE on Health Research that continue to view the issues raised in this paper as problematic[40 41] and practitioner calls for greater investment in research capacity building and its evaluation to support emerging research agendas.[21 42]

## CONCLUSION

Barriers and enablers to locally led trial undertaking exist at all levels and functions of LMIC research systems. Establishing the necessary conditions to facilitate this research will require multiple, coordinated interventions that seek to resolve them in a systemic manner. The conceptual framework and strategies presented in this paper provide an evidence-based framework for implementing a self-sustaining capacity development approach. This guidance is relevant for policy makers and funders and local and international researchers who have a critical responsibility for ensuring their research efforts are dedicated to developing the systems in which they work.

**Author affiliations**
[1]The Global Health Network, Centre for Tropical Medicine and Global Health, University of Oxford, Oxford, UK
[2]Oxford Policy Management, Oxford, UK
[3]Department of Global Health and Development, London School of Hygiene & Tropical Medicine, London, UK
[4]Department of Medicine, Faculty of Medicine & Allied Sciences, Rajarata University of Sri Lanka, Saliyapura, Sri Lanka
[5]Department of Public Health and Hygiene, Faculty of Health Sciences, University of Buea, Buea, Cameroon
[6]Centre for Clinical Vaccinology & Tropical Medicine, University of Oxford, Oxford, UK

**Acknowledgements** This article is dedicated to the memory of one of the authors, Dr Julius Atashili, who sadly passed away before this article was published. His commitment and contribution to research and teaching will be remembered by all those who were fortunate enough to know him as a colleague or mentor. The authors would also like to thank all the individuals and institutions that participated in and facilitated the case studies in Ethiopia, Cameroon and Sri Lanka.

**Contributors** SRPF conceived, designed and implemented the study, analysed the data and drafted the manuscript with input and assistance from CC, BA and TL. SS and JA collaborated on the design, implementation, interpretation of data and critically revised and reviewed versions of the manuscript. All authors, except JA, approved the final version of the manuscript.

**Funding** This work was supported by The Global Health Network, a Bill and Melinda Gates Foundation funded project (grant reference: OPP1053843), and in part by The Medical Research Council and Nuffield Department of Medicine Doctoral Prize Studentship. The funders had no role in study sponsorship.

**Competing interests** None declared.

**Ethics approval** The study was approved by the University of Oxford Tropical Research Ethics Committee (OXTREC Reference 70-11); Ministry of Health, Cameroon (Ref: 631-07-12); University of Yaoundé, Faculty of Medicine and Biomedical Sciences (Ref: 0694); University of Buea, Faculty of Health Sciences (Ref: 2011-12-0041); and University of Sri Jayewardenepura, Faculty of Medical Sciences (Application No: 636/12).

**Provenance and peer review** Not commissioned; externally peer reviewed.

**Data sharing statement** The research protocol, methodology and case-specific data and reports are available online at: https://globalhealthtrials.tghn.org/articles/strategies-developing-sustainable-health-research-capacity-low-and-middle-income-countries/.

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
