## [Reviewer comments · BMJ Open]

ARTICLE DETAILS

TITLE (PROVISIONAL)	Strategies for developing sustainable health research capacity in Low and Middle Income Countries; a prospective, qualitative, multi-site study investigating the barriers and enablers to locally-led clinical trial conduct in Ethiopia, Cameroon, and Sri Lanka
AUTHORS	Franzen, Samuel; Chandler, Clare; Siribaddana, Sisira; Atashili, Julius; Angus, Brian; Lang, Trudie

VERSION 1 - REVIEW

REVIEWER	Dr Marguerite Schneider Alan J Flisher Centre for Public Mental Health University of Cape Town South Africa
REVIEW RETURNED	15-May-2017

GENERAL COMMENTS	General comments This is a well-written paper presenting an interesting and well-conceptualised study. The use of the framework that reviews individual, operational and macro/institutional elements that will determine the feasibility of locally-led trials to be conducted provides a comprehensive perspective on the barriers and facilitators to developing trial capacity in LMICs. The presentation of the overall findings in Table 2 is particularly good as it provides a good plan of action template to follow in building capacity in LMICS. There are a few gaps in the methodology section that need addressing. The themes are clearly identified and seem valid and efficient in elucidating the framework used. It would be useful and interesting to get some sense of what branches of health research were included – e.g. basic drug trials, more pragmatic research including implementation research, etc. Abstract and Introduction I have no specific comments on these sections. They are clear and concise. Methodology P5 line 15: change 'heath' to 'health' P5 line 24: 'According to their profile, participants were selected to take part in interviews, focus groups, or process mapping exercises.' Please give some indications of criteria used for this selection process. P5 line 34 – 41: What about ethics approval in Ethiopia? P6 line 4: it would be useful to have a brief overview of the Ethiopian case study especially where it differed from the other 2. Results P7 line 46-47: 'Findings from the systematic review are also compared for reference later in the discussion.' Make it clear here that it is the previous systematic review conducted by the authors
---

	and not part of the methods for this paper. P7 lines 51-52: 'Rather, the differential existence of barriers and enablers made them more or less influential on trial conduct in particular settings.' This sentence is difficult to understand. What is a 'differential existence'? Rephrase it into a simpler sentence. P7 lines 53-54: '...these transferable factors.' What are these factors? P11 quote starting at line 14: seems a very long quote. It can be reduced if word count is problematic. Discussion and conclusions The differences and similarities between the 3 case studies could be explored more. The material in Table 2 is great but it would be good to try and identify some key issues and strategies to start with rather than being overwhelmed with all the requirements set out in the table. Style, formatting and editorial care Table 2: Why is some text in italics? If not a formatting error please indicate the meaning. Also make sure that the headings appear on each individual page to facilitate the reading of the table without having to scroll up all the time. Fig 3 is very small print. Can a higher font size be used? Generally the style of writing and attention paid to formatting and editorial care are commendable. There are a few instances where the language is overly complex and I have pointed these out above.
--	---

REVIEWER	Scott McIntosh University of Rochester Medical Center Rochester, NY, USA
REVIEW RETURNED	17-May-2017

GENERAL COMMENTS	This is a well written report of an interesting qualitative and case study approach in 3 countries. The authors are mindful of the study's limitations and transparent about all methods. Their developed conceptual model is intriguing and will be of interest to global health researchers. Major Comments: No major comments. Minor Comments: The title is very long, so authors should consider shortening if possible (optional). It appears the headings go from 3.1 to 3.2, then start again with 3.1 for Individual level. This was confusing.
--

VERSION 1 – AUTHOR RESPONSE

Reviewer: 1

Reviewer Name: Dr Marguerite Schneider

Institution and Country: Alan J Flisher Centre for Public Mental Health, University of Cape Town, South Africa
Competing Interests: I have no competing interests

General comments

This is a well-written paper presenting an interesting and well-conceptualised study. The use of the framework that reviews individual, operational and macro/institutional elements that will determine the

feasibility of locally-led trials to be conducted provides a comprehensive perspective on the barriers and facilitators to developing trial capacity in LMICs. The presentation of the overall findings in Table 2 is particularly good as it provides a good plan of action template to follow in building capacity in LMICS.

Thank you. We appreciate your positive comments.

There are a few gaps in the methodology section that need addressing.

The themes are clearly identified and seem valid and efficient in elucidating the framework used.

It would be useful and interesting to get some sense of what branches of health research were included – e.g. basic drug trials, more pragmatic research including implementation research, etc. The following paragraph has been added to the study population and research context section of the results (section 3.1, line 171):

“In all case study countries, most research was pre-clinical, using descriptive designs such as case studies or cross sectional analysis. Of the experimental research done, only a small proportion were clinical trials. The clinical trials conducted by participants mostly investigated the use of previously approved therapeutics to improve the treatment of locally-important diseases, with a few investigating operational topics including behavioural interventions (for instance, to improve drug adherence). Investigation of novel therapeutics was rare, although in Sri Lanka the use of clinical trials to determine the effectiveness of traditional medicines was common.”

Abstract and Introduction

I have no specific comments on these sections. They are clear and concise.

Methodology

P5 line 15: change 'heath' to 'health'

Corrected

P5 line 24: 'According to their profile, participants were selected to take part in interviews, focus groups, or process mapping exercises.' Please give some indications of criteria used for this selection process.

Added to row 135:

“Interviews were used to explore management, governance and other sensitive issues which would not be appropriate for group discussion, and when scheduling difficulties meant that group discussions were not possible. Focus groups were conducted with participants who had a variety of research experiences, in order to explore a breadth of perspectives. Process mapping exercises were used with specific research teams who had recently conducted a clinical trial. The purpose of this exercise was to systematically walk through and map the process of conducting a clinical trial.”

P5 line 34 – 41: What about ethics approval in Ethiopia?

The case study in Ethiopia began as an informal pilot, to generate ideas, questions and guide planning for a subsequent larger study. OXTREC approval was gained through an expedited review process because it was classified as minimal risk. We did not apply for Ethiopian ethics approval because of the exploratory and minimal risk nature of the work. However, during the pilot work in Ethiopia, it became clear that useful information was being generated and the research should be extended. The local researchers we worked with informed us that this would be appropriate without Ethiopian ethics approval, if each participant was fully informed that the study only had Oxford approval, and still agreed to take part. After verbal consent, each participant was subsequently

contacted to explain this and to gain written confirmation of consent.

P6 line 4: it would be useful to have a brief overview of the Ethiopian case study especially where it differed from the other 2.

The authors feel that significant further detail on the Ethiopian case study would add little to the paper.

A comparison on the methodological differences is presented in the methods section:

- Within Figure 1
- Line 118: "Accordingly, the first case study in Ethiopia was designed as a pilot to explore issues affecting locally led trial conduct and develop a preliminary conceptual framework."
- Line 153: "In Ethiopia, participants said that they would be more comfortable giving verbal informed consent, and not being audio recorded. Accordingly, detailed notes were taken with quotes noted as near verbatim as possible, detailing identification numbers."
- Reference to the full article on the Ethiopian case study are presented in line 173

However, we realise that implications on data saturation were not sufficiently clear, so we have added the following sentence to line 149: "However, due to the smaller sample size in the Ethiopian pilot study, saturation of themes was not apparent."

Results

P7 line 46-47: 'Findings from the systematic review are also compared for reference later in the discussion.' Make it clear here that it is the previous systematic review conducted by the authors and not part of the methods for this paper.

Added to line 125 "A parallel systematic review on health research capacity development was also conducted to determine if case-study findings were more widely generalizable. The methods and findings of this review are published as a separate article²⁴"

Added to line 200: "Findings from the systematic review published as a separate article²⁴ are also compared for reference later in the discussion."

P7 lines 51-52: 'Rather, the differential existence of barriers and enablers made them more or less influential on trial conduct in particular settings.' This sentence is difficult to understand. What is a 'differential existence'? Rephrase it into a simpler sentence.

P7 lines 53-54: '...these transferable factors.' What are these factors?

Rephrased paragraph (line 204) to address above 2 point: "Although some barriers and enablers were more or less influential in different research contexts, the majority were present in every case study. There were no contradictory findings whereby specific barriers and enablers were not considered important. A conceptual model of the necessary conditions for locally led trial conduct was developed to demonstrate the interaction between these common barriers and enablers."

P11 quote starting at line 14: seems a very long quote. It can be reduced if word count is problematic.
Row 350: Cut down

Discussion and conclusions

The differences and similarities between the 3 case studies could be explored more.

Paragraphs added to summary section (from row 573):

A detailed comparison of the barriers and enablers identified in the three case is presented in

Supplementary File 1. This comparison and the conceptual model (Figure 3) suggests that Sri Lanka was more productive in terms of its clinical trial and research outputs compared to Ethiopia and Cameroon due to an enabling research environment that can be traced back to pro-research cultures at multiple levels. Resources for research were more available within Sri Lanka in terms of national grants, better quality infrastructure and equipment, and stronger incentives for conducting research, at least within academia. These basic pre-requisites supported locally-led trials and meant that local researchers were not dependent on international collaboration or parallel research structures. The resulting higher volume of national research is, in turn, likely to explain why networking, exposure to clinical trials, and self-efficacy were less problematic in Sri Lanka. Finally, the availability of locally generated evidence appeared to meet most of the needs of policymakers because a preference for international evidence was not mentioned. This may explain the greater buy-in for research at policy level, which is evidence by greater research investments.

However, while this was true for academia, the problems facing research in healthcare environments in Sri Lanka were similar to those in Ethiopia and Cameroon. Stewardship and governance capacity, and the availability of human resources capable of conducting clinical trials, were also limiting factors in all countries. This suggests that while specific national investments can be helpful, a whole-of-systems approach is needed to comprehensively address the issues facing locally-led research in developing countries.”

The material in Table 2 is great but it would be good to try and identify some key issues and strategies to start with rather than being overwhelmed with all the requirements set out in the table.

Added the following paragraph after line 566:

Strategies under “Fostering pro-research cultures” focus on generating top-level buy-in to secure investment, generate support and appreciation for trial research and increasing the pool of potential researchers willing and confident enough to conduct trials. “Developing trial leaders and staff” concentrates on resolving skills gaps of academics and healthcare staff so that they can undertake trials, and developing future research leaders that have the capabilities to successfully manage trials in challenging environments, support the development of local staff and institutions, and can act as champions for change. “Providing a facilitative operational environment for trials” aims to reduce operational barriers to trial conduct and increase material resources so that future trials can be conducted with greater scope, quality and ease, therefore making trial conduct within local institutions a more attractive option. “Ensuring trial research has an impact” not only aims to make clinical trial evidence useful for policy, but also to demonstrate that local research is credible, valuable, and offers a good return on investment so that pro-research cultures and support for trials is reinforced.”

Style, formatting and editorial care

Table 2: Why is some text in italics? If not a formatting error please indicate the meaning. Also make sure that the headings appear on each individual page to facilitate the reading of the table without having to scroll up all the time.

Italics used in table 2 to indicate reference to another group of strategies shown previously in the table. However, we appreciate this is confusing and unnecessary, so removed italics, and instead provided direction to relevant sections in brackets i.e. (see section above/below).

Added header rows to each page of table 2.

Fig 3 is very small print. Can a higher font size be used?

We managed to increase it by 2 points but it is very difficult to fit in any larger text. When printed the

image can be blown up larger. If the editors have any guidance, it would be appreciated.

Generally the style of writing and attention paid to formatting and editorial care are commendable. There are a few instances where the language is overly complex and I have pointed these out above. Thanks. These comments have been addressed above.

Reviewer: 2

Reviewer Name: Scott McIntosh

Institution and Country: University of Rochester Medical Center, Rochester, NY, USA

Competing Interests: None declared.

This is a well written report of an interesting qualitative and case study approach in 3 countries. The authors are mindful of the study's limitations and transparent about all methods. Their developed conceptual model is intriguing and will be of interest to global health researchers.

Thank you.

Major Comments: No major comments.

Minor Comments:

The title is very long, so authors should consider shortening if possible (optional).

Thank you. We appreciate that it is a bit clunky and not very catchy, but we followed the BMJ guidance for titles which includes the study design and description of phenomenon of study. Also we feel that the level of information in the title will help it get identified on searches.

It appears the headings go from 3.1 to 3.2, then start again with 3.1 for Individual level. This was confusing.

Apologies. Corrected.

VERSION 2 – REVIEW

REVIEWER	Dr Marguerite Schneider Alan J Flisher Centre for Public Mental Health University of Cape Town South Africa
REVIEW RETURNED	23-Jun-2017

GENERAL COMMENTS	The revisions undertaken have addressed my earlier concerns effectively. The additions are well presented. There are no further comments requiring revisions. There is 1 typographical error that I noticed - 'literature' in the discussion.
---